# One-shot Federated Learning with Training-Free Client

## Abstract

While traditional iterative federated learning (FL) is often limited by various factors, such as massive communication overhead, higher risk of being attacked, and fault tolerance requirements, an emerging and promising solution is to conduct FL with a single communication round, termed one-shot FL. However, a lack of continuous communication leads to the serious performance degradation of current FL frameworks, especially training with statistical heterogeneity, *i.e.*, Non-IID. The primary objective of this paper is to develop an effective and efficient one-shot FL framework to better deal with statistical heterogeneity. To achieve this, we first revisit the influence of statistical heterogeneity on model optimization and observe that conventional mechanism (*i.e.*, training from scratch and parameter averaging) is inadvisable for one-shot FL due to the problem of client drift. Based on this observation, we propose a novel one-shot FL framework, namely FedTC. Different from existing methods, FedTC divides the model into backbone and head, deploying them separately on the client and server sides. Specifically, our approach does not directly train the whole model on biased local datasets from scratch, but only learns a detached head through unbiased class prototypes estimated by the pre-trained backbone. Moreover, we integrate the feature outlier filtering strategy and adapter into our FedTC to further improve its performance. Extensive experiments demonstrate that FedTC can significantly outperform several state-of-the-art one-shot FL approaches with extremely low communication and computation costs.

## 1 Introduction

Federated learning (FL) (Li et al., 2020b; McMahan et al., 2017; Yang et al., 2019) collaboratively trains a global model through information exchange between clients and the central server without compromising individual privacy. It has recently gained increasing attention from the community as a powerful tool for distributed learning, particularly in privacy-sensitive scenarios like healthcare (Feng et al., 2022; Liu et al., 2021; Karargyris et al., 2023). Nevertheless, the traditional FL paradigm requires numerous rounds of continuous communication, leading to substantial communication overhead and a high risk of being vulnerable to attacks. Additionally, it imposes high requirements on communication conditions and fault tolerance.

One-shot FL (Guha et al., 2019) is an emerging concept aimed at addressing the aforementioned challenges by communicating in ***just one round***. However, data coming from discrete clients are inevitably diverse (*i.e.*, Non-IID) (Zhao et al., 2018) in real-world applications, *e.g.*, label distribution skew (Li et al., 2022). The substantial statistical heterogeneity arising from variations in underlying distributions among clients leads to a divergence in local optimization, known as ***client drift*** (Karimireddy et al., 2020). To tackle this issue, previous studies have introduced a range of techniques, *e.g.*, regularization (Li et al., 2020c; 2021b), distillation (Lee et al., 2022; Yao et al., 2021), and normalization (Wang et al., 2020), aiming at revising the processes of local training and parameter averaging of FedAvg. However, their performance may significantly deteriorate if we simply decrease the number of communication round to one. The underlying reason for the degradation is that these techniques heavily rely on multiple rounds of communication to gradually train and tune the models. Therefore, the issue of statistical heterogeneity is a fundamental challenge in one-shot FL.

Existing one-shot FL methods can be roughly divided into two categories: ***ensemble-based*** (Su et al., 2023; Diao et al., 2023; Guha et al., 2019) and ***distillation-based*** (Li et al., 2020a; Lin et al., 2020; Zhang et al., 2022). The former one directly ensembles the output of local models as the final prediction, while the latter one requires an additional distillation step on the server side. Both of them share some practical limitations. On the one hand, neither of them has addressed the problem of client drift, and instead directly utilizes biased local models for voting or distillation, which is a sub-optimal solution. On the other hand, the existing ensemble-based/distillation-based methods unavoidably increase the computational overhead and may raise some demanding requirements like additional datasets. In addition, they still require training procedures on the client side, which may be difficult to implement on edge devices, *i.e.*, ***Edge AI*** (Lim et al., 2020; Nguyen et al., 2021). In fact, the clients in FL are usually edge devices, such as mobile phones and wearable devices with limited computational capacity. This motivates us to develop an efficient framework to implement one-shot FL.

In this work, we first conduct an in-depth analysis of the impact of statistical heterogeneity on the backbone and head of the model, which are responsible for feature extraction and model predictions, respectively. From the analysis, we have obtained two important observations: ❶ Training model from scratch will lead to significant drift on the backbone and head. ❷ Averaging the parameters of local models trained from one-shot FL is infeasible due to serious client drift. Based on the analysis, we propose a novel one-shot **Fed**erated learning framework with **T**raining-free **C**lients, namely FedTC, to effectively address the client drift problem in a single round.

FedTC is a divide-and-conquer framework, which decomposes the client drift problem into two sub-problems, *i.e.*, backbone drift and head drift. To tackle the issue of backbone drift, our FedTC utilizes pre-trained weights to initialize the backbone and conducts the forward process to estimate the class prototypes. Without model updates on biased datasets, the proposed FedTC thoroughly addresses backbone drift. Then, the prototypes from each client are collected to create an unbiased prototype dataset, which is used to train a detached head on the server side. Such a strategy can effectively address the issue of head drift. However, there still exists a gap between the pre-trained dataset and the real-world dataset. To bridge this gap, our FedTC adds an adapter to the head to learn domain-specific knowledge from prototypes on the server. On the client side, a feature outlier filtering strategy is utilized to remove outliers effectively. Extensive experiments on five benchmarks demonstrate that the proposed FedTC significantly improves accuracy compared to existing one-shot FL approaches under various data heterogeneity settings, and dramatically reduces communication and computation costs in the meantime. Furthermore, our method excludes the training on the client side, which increases its potential for the deployment in Edge AI applications.

## 2 RELATED WORK

**One-shot Federated Learning.** While iterative federated learning methods (McMahan et al., 2017; Li et al., 2021b; 2022; 2020c; Karimireddy et al., 2020; Wang et al., 2020; Lee et al., 2022) have achieved success, they are still constrained by the significant communication and computation costs due to multiple rounds of training. One-shot FL presents a promising solution to this issue, initially proposed by (Guha et al., 2019). The objective is to accomplish efficient federated learning within a single round. However, due to the presence of statistical heterogeneity, *i.e.*, Non-IID data, in real-world applications, a straightforward way that changing the number of rounds to one for iterative FL methods would result in considerably deteriorated model performance. Consequently, addressing statistical heterogeneity in one-shot FL has emerged as an important research direction. To handle this challenge, researchers distill the biased knowledge (Lin et al., 2020; Li et al., 2020a; Zhang et al., 2022) of local models to the global model or directly ensemble them to make the final prediction (Guha et al., 2019; Diao et al., 2023; Su et al., 2023). However, they did not address the model bias but instead circumvented the impact of model bias through distillation or ensemble, which is a sub-optimal solution. In contrast to these methods, our approach effectively deals with the detrimental effects of client drift by simultaneously addressing issues stemming from both the backbone and the head aspects of the model.

**Pre-trained Foundation Model.** As model parameters and available data continue to grow, the utilization of foundational models such as ViT (Dosovitskiy et al., 2020), BERT (Devlin et al., 2018), and GPT-4 (OpenAI, 2023), which have been pre-trained on large datasets to capture abun-

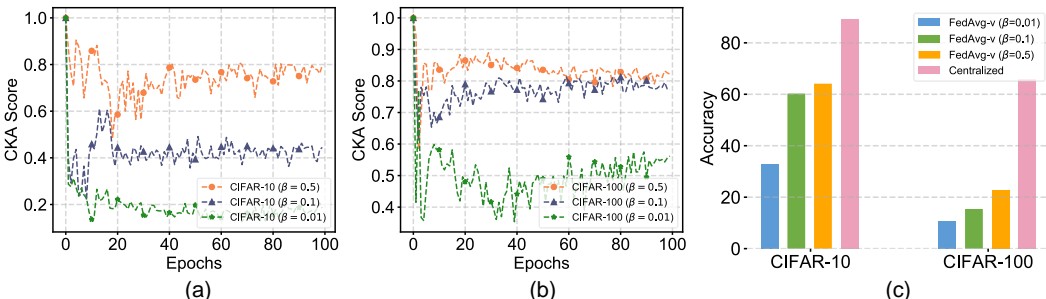

Figure 1: **Analysis of the impact of statistical heterogeneity** on both backbone and head. We illustrate the CKA (Kornblith et al., 2019) score between two different local models versus local epochs on (a) **CIFAR-10** and (b) **CIFAR-100** (Krizhevsky et al., 2009) under various non-IID settings. (c) The test accuracy (%) comparison between centralized learning and FedAvg-v.

dant knowledge, has become a valuable approach for enhancing downstream tasks like object detection (Joseph et al., 2021; Dai et al., 2021), semantic segmentation (Xu et al., 2022), and visual question answering (Li et al., 2023). The success of this strategy has also garnered attention in the field of FL. Leveraging pre-trained models presents the opportunity to significantly reduce the communication and computational costs associated with training larger models (Tian et al., 2022; Hamer et al., 2020). Furthermore, some researchers (Chen et al., 2023a; Tan et al., 2022b) have attempted to incorporate pre-trained models to alleviate model bias caused by statistical heterogeneity in iterative FL. However, we surprisingly find that little attention is paid to how to effectively apply pre-trained models in one-shot FL.

**Prototype Learning.** A prototype is defined as the mean feature vector within a class, subtly encapsulating the notion of clustering. It is a popular and effective method widely used in various tasks (Gao et al., 2021b; Yang et al., 2018; Li et al., 2021a; Deng et al., 2021) to perceive class-specific information. In iterative federated learning, prototype learning has been introduced in various studies as a form of regulation to guide local training (Huang et al., 2023; Tan et al., 2022b;a). However, it may not be directly applicable to one-shot federated learning scenarios.

## 3 METHODOLOGY

### 3.1 PRELIMINARY: ONE-SHOT FEDERATED LEARNING

Assuming a federation consists of $K$ clients and a single trusted server assigned for communication. Each individual client, denoted as $C^k$ ($k \in [K]$), possesses its own private dataset $D^k$, which comprises a set of $n^k$ training instances $\{X_i^k, Y_i^k\}_{i=1}^{n^k}$, where $X$ representing the images and $Y \in V$ representing the corresponding labels; $V$ is the label space of the whole federation. Note that the distribution of these local datasets is skewed. Besides, each client has a deep neural network $f(w_{\mathcal{B}}^k, w_{\mathcal{H}}^k) \triangleq \mathcal{B}(w_{\mathcal{B}}^k) \circ \mathcal{H}(w_{\mathcal{H}}^k)$, where $\mathcal{B} : X \to Z$ is the backbone to extract features $Z$ from images, and $\mathcal{H} : Z \to Y$ is the head to make predictions. The primary objective of one-shot FL is to achieve the optimal global model parameters $w_{\mathcal{B}}^* \circ w_{\mathcal{H}}^*$ within a single communication round.

### 3.2 MOTIVATION

Data heterogeneity poses a fundamental challenge in FL, particularly with the constraints of the one-shot setting. Most traditional FL methods adhere to the parameter averaging mechanism of FedAvg in the server. However, such a mechanism may lead to a global model with significantly low accuracy on clients under the one-shot setting. To this end, we undertake a comprehensive analysis of how data heterogeneity impacts the optimization of models. Particularly, we disassemble the model into its backbone and head components, thus enabling an isolated investigation. In previous studies, these constituents (*i.e.*, backbone and head) have been treated as an indivisible whole. However, they may exhibit varying behaviors concerning data heterogeneity, due to their distinct responsibilities.

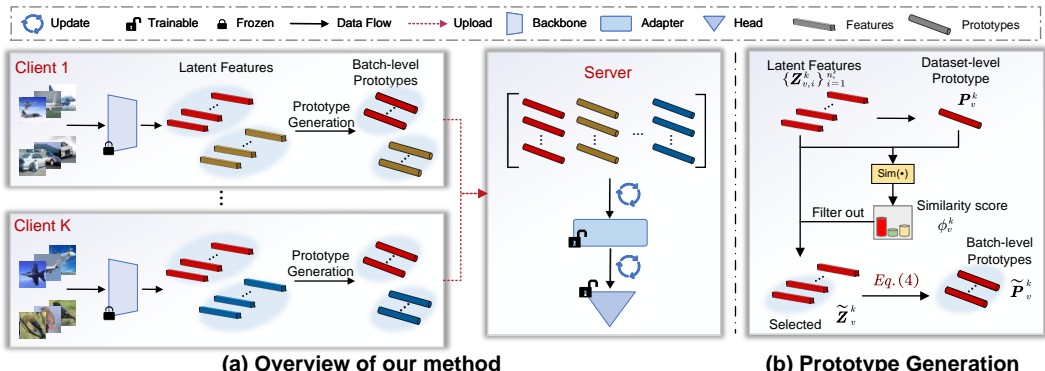

Figure 2: **Illustration of FedTC.** (a) Overview of the proposed method. (b) The detail of prototype generation in client-side execution. *On the client side*, we first utilize the fixed pre-trained backbone to infer latent features per category (coloured by different colours), with outliers being filtered out through the similarity between them and the dataset-level prototype. The carefully selected features are used to calculate batch-level prototypes according to Eq.(4), which will be transferred to the server. *On the server side*, we aggregate the prototypes from the clients and employ them to train the head. Besides, we leverage an adapter to further mitigate the domain shift between pre-trained and real-world datasets. Zoom in for details.

We commenced by initializing the local models with an identical pre-trained backbone for FedAvg. Subsequently, we visualized the discrepancy versus local epochs in the backbones between the two clients on CIFAR-10 and CIFAR-100 (Krizhevsky et al., 2009). This visualization, as depicted in Fig. 1(a) and (b), was conducted using the Linear CKA (Kornblith et al., 2019), which serves as an indicator of the similarity in feature representations. As shown, updating local models on biased datasets results in significant representation differences, and is more pronounced as data heterogeneity increases, *i.e.*, backbone drift. Hence, the global backbones obtained from averaged parameters of local backbones have terrible feature representation ability. In this regard, we argue that *freezing the backbone is a better choice*, even when a potential domain shift exists between pre-training and real datasets. This is because the primary factor leading to model degradation is client drift caused by statistical heterogeneity.

In order to delve deeper into the impact of statistical heterogeneity on the head, we constructed a variant of FedAvg, namely FedAvg-v, which has a frozen and pre-trained backbone and conducts parameter averaging only on the local heads. We compare it with the centralized learning method, *i.e.*, changing the number of clients to 1 for FedAvg-v. Illustrated in Fig. 1(c), a substantial gap emerges between these two approaches and increases as larger heterogeneity. This indicates that statistical heterogeneity leads to biased local heads, causing a notable degradation in performance, *i.e.*, head drift. Therefore, we further argue that *the parameter averaging mechanism is infeasible in one-shot FL*, even when only applied to the head, which consists of a relatively small number of parameters. Motivated by prototype learning, a recently popular solution in FL (Tan et al., 2022a;b; Huang et al., 2023), we proposed a detached head in the server which trained on unbiased prototypes to address the head drift.

## 3.3 FedTC

In this section, we proposed a novel one-shot FL framework named FedTC based on the above insights. To address the drift in both the backbone and head, FedTC proposes a separate network architecture that a frozen and pre-trained backbone on the client side and a trainable head on the server side. Additionally, it incorporates a feature outlier filtering strategy and an adapter to mitigate the domain shift between the pre-trained dataset and the real-world dataset. We present the overview of FedTC in Fig. 2 and detailed algorithm in Appendix A.1. The subsequent parts elaborate on the details of our proposed method.

### 3.3.1 CLIENT-SIDE EXECUTION

Different from previous work, there is no training procedure in the client of FedTC, instead of leveraging the pre-trained backbone to estimate the class prototypes of images. Specifically, for each image $\boldsymbol{X}_i^k \in \mathbb{R}^{\mathcal{C} \times H \times W}$ from local dataset $D^k$, with spatial size $(H \times W)$ and $\mathcal{C}$ channels, we can get the latent feature vector $\boldsymbol{Z}_i^k = \mathcal{B}^k(\boldsymbol{w}_{\mathcal{B}}, \boldsymbol{X}_i^k), \in \mathbb{R}^L$ by feeding it into the backbone, where $L$ is the dimension of the feature vector and $\boldsymbol{w}_{\mathcal{B}}$ is the pre-trained parameters.

**Feature Outlier Filtering.** Due to the domain shift between the pre-trained and real-world datasets, it is inevitable that the latent features $\{\boldsymbol{Z}_i^k\}_{i=1}^{n^k}$ contains the outliers, thereby decreasing the quality of prototypes. To tackle this issue, we propose a filtering strategy based on the ranking of scores, quantified by the distance between features and the class center. Firstly, we can estimate the center of the class $v$ belongs to $C^k$ by dataset-level prototype $\boldsymbol{P}_v^k$ as follows:

$$\boldsymbol{P}_v^k = \frac{1}{n_v^k} \sum_{i=1}^{n_v^k} \boldsymbol{Z}_{v,i}^k, \quad \boldsymbol{P}_v^k \in \mathbb{R}^L, \quad \text{where } 0 < n_v^k \leq n^k \text{ and } v \in \boldsymbol{V}. \tag{1}$$

Then, we utilize the prototype as the anchor and measure the distance between each latent feature and the anchor by cosine similarity:

$$\phi_v^k = [\phi_{v,1}^k, \ldots, \phi_{v,i}^k, \ldots, \phi_{v,n_v^k}^k], \quad \phi_{v,i}^k = \frac{\boldsymbol{Z}_{v,i}^k \cdot \boldsymbol{P}_v^{k^T}}{\|\boldsymbol{Z}_{v,i}^k\| \times \|\boldsymbol{P}_v^k\|}. \tag{2}$$

The similarity score $\phi_v^k \in \mathbb{R}^{n_v^k}$ signifies the distance between each latent feature and the dataset-level prototype within the samples belonging to class $v$. As outliers typically exhibit a greater separation from the anchor, we can effectively identify and filter them out based on the ranking of similarity score:

$$\widetilde{\boldsymbol{Z}}_v^k = \{\boldsymbol{Z}_{v,i}^k\}_{i \in \mathcal{Q}}, \quad \text{and } \mathcal{Q} = \mathbb{I}(top(\phi_v^k, \alpha)), \quad \alpha \in (0, 1], \tag{3}$$

where $top$ refers to selecting the top $\alpha n_v^k$ features with the highest similarity scores and $\mathbb{I}$ denotes the operation of accessing their indices $\mathcal{Q}$.

**Prototypes Estimation.** After filtering out the feature outliers for each class, we can estimate the batch-level prototypes within a random $B$ mini-batch latent features, which can be described as follows:

$$\widetilde{\boldsymbol{P}}_v^k = [\ldots, \widetilde{\boldsymbol{P}}_{v,j}^k, \ldots], \quad \widetilde{\boldsymbol{P}}_{v,j}^k = \frac{1}{B} \mathbb{E}_{B_j \sim \mathcal{Q}} \sum_{i \in B_j} \widetilde{\boldsymbol{Z}}_{v,i}^k, \quad \text{and } 0 < j \leq \frac{\alpha n_v^k}{B}. \tag{4}$$

Prototypes from all classes within the clients are transmitted to the server for training.

### 3.3.2 SERVER-SIDE EXECUTION

On the server side, we gather all the local prototypes to construct a prototype dataset $\widetilde{D} = \{\widetilde{\boldsymbol{P}}_v, v\}_{v \in \boldsymbol{V}}$, where $\widetilde{\boldsymbol{P}}_v = \{\widetilde{\boldsymbol{P}}_v^k\}_{k \in \mathcal{K}_v}$ and $\mathcal{K}_v$ denotes the indices of the client who has class $v$ samples. These prototypes contain specific knowledge for each class, which contributes to learning the decision boundary and subsequently making accurate classifications. Notably, in contrast to the local datasets, the prototype dataset is unbiased as it aggregates prototypes from all classes. Consequently, a model trained on this dataset will remain unaffected by statistical heterogeneity. However, owing to domain shift, the pre-trained backbone may possess a limited representation capacity in the practical dataset, resulting in the head not being able to effectively recognize different classes. Although fine-tuning the backbone directly on local datasets might seem like an option, it could compromise the rich knowledge acquired from the extensive dataset and lead to backbone drift. Inspired by the visual adapters (Chen et al., 2023b; Wu et al., 2023; Chen et al., 2022; Gao et al., 2021a), a recent and noteworthy technique has demonstrated the capability to strike a harmonious equilibrium between acquiring domain-specific knowledge and retaining the extensive knowledge embedded in the pre-trained model. We incorporate this technique to tackle the aforementioned challenge. The adapter in FedTC has a similar architecture to the projector in FedPCL (Tan et al., 2022b), containing multiple continuous FC-Relu-Norm operations, as the detailed structure is provided in Appendix A.2. One difference is that FedPCL introduced its projector to align the feature dimensions from different networks. Another significant distinction lies in FedPCL putting the projector into local models, which

is applicable in iterative federated learning but impractical in one-shot FL due to backbone drift. In contrast, we introduced the adapter ahead of the detached head in the server to enable the acquisition of domain-specific knowledge from prototypes. This helps the head to learn better decision boundaries for different classes. The loss function can be described as:

$$\mathcal{L} = \sum_{(\widetilde{\boldsymbol{P}}_i, v_i) \in \widetilde{D}} l_{ce}(\mathcal{G}(\boldsymbol{w}_{\mathcal{G}}, \mathcal{H}(\boldsymbol{w}_{\mathcal{H}}, \widetilde{\boldsymbol{P}}_i)), v_i), \tag{5}$$

where $\mathcal{G}$ is the adapter with the parameter $\boldsymbol{w}_{\mathcal{G}}$ and $l_{ce}$ is the cross entropy loss function.

### 3.3.3 PRIVACY SECURITY

Transmitting prototypes is a common practice in FL (Tan et al., 2022b;a; Huang et al., 2023), as prototypes are statistical-level information of class that does not contain the privacy of individual samples. In Appendix C, we provide a comprehensive analysis of the privacy security for prototypes, which was missing in prior works. The analysis results demonstrated that our approach ensures privacy preservation and adheres to the rules of FL.

## 4 EXPERIMENTS

### 4.1 EXPERIMENTAL SETUP

**Datasets.** We conduct extensive experiments on five widely-used benchmarks: **MNIST** (LeCun et al., 1998), **Fasion-MNIST** (Xiao et al., 2017), **CIFAR-10** (Krizhevsky et al., 2009), **CIFAR-100** (Krizhevsky et al., 2009) and **Mini-ImageNet** (Vinyals et al., 2016; Dong et al., 2022). Following the previous work (Diao et al., 2023), we divided the data into 10 clients with an activity ratio of 1 by two different Non-IID partition strategies:

- **Sharding** (McMahan et al., 2017; Diao et al., 2023): For datasets with fewer classes (e.g., 10 for MNIST, Fashion-MNIST, and CIFAR-10), we randomly assigned $\delta \in \{1, 2, 3\}$ classes to each client, ensuring an equal sample count for every class.
- **LDA** (Li et al., 2021b; Diao et al., 2023; Zhang et al., 2022): We employ the Latent Dirichlet Allocation (LDA) strategy to partition the data among each client. In this approach, each local dataset is sampled from the Dirichlet distribution $Dir(\beta)$, where a larger $\beta \in \{0.01, 0.1, 0.5\}$ corresponds to smaller data heterogeneity.

**Baselines.** We compared our method with several state-of-the-art one-shot federated learning methods. This comparison included two distillation-based methods, *i.e.*, **FedDF** (Lin et al., 2020) and **DENSE** (Zhang et al., 2022), as well as one ensemble-based approach, *i.e.*, **FedOV** (Diao et al., 2023). Additionally, we included several typical iterative FL methods such as **FedAvg** (McMahan et al., 2017), **FedProx** (Li et al., 2020c), and **MOON** (Li et al., 2021b) as baselines. To ensure fairness, we executed these comparisons within a single round. The performance is quantified using the top-1 accuracy.

**Network Architectures.** We utilize the ResNet-50 (He et al., 2016) architecture with pre-trained weights from CLIP (Radford et al., 2021) for CIFAR-10, CIFAR-100, and Mini-ImageNet. For MNIST and Fashion-MNIST, we employ the ResNet-50 weights pre-trained from the ImageNet (Deng et al., 2009) dataset. For a fair comparison, all methods employ the aforementioned network architecture, and the adapter is also integrated. However, we were surprised to find that some methods exhibited a lack of robustness concerning the network structure. Therefore, we include the additional version of all baselines that utilize a SimpleCNN (Diao et al., 2023; Zhang et al., 2022) network in their paper.

**Implementation Details.** We implemented all methods using PyTorch and conducted the training on a single RTX 3090 GPU with 24GB of memory. Consistent with the setting in FedOV (Diao et al., 2023), we employed the SGD optimizer with a learning rate of 0.001 and a batch size of 64. For the local training phase of all baseline methods, we set the local epoch to 200. Similarly, the training at the server was carried out for 200 epochs. Our hyper-parameters $\alpha$ and $B$ are set to 0.99 and 5 as default. Due to the page limitation, we present the additional details and results of our experiments in Appendix B.

Table 1: **Test accuracy (%) of all approaches on MNIST, Fashion-MNIST, CIFAR-10, CIFAR-100, and Mini-ImageNet** under two different non-IID settins. The best results are marked in bold.

| | | **Non-IID Partition Strategy: Sharding** | | | | | | | | |
|---|---|---|---|---|---|---|---|---|---|---|
| **Method** | **Model** | **MNIST** | | | **Fasion-MNIST** | | | **CIFAR-10** | | |
| | | $\delta=1$ | $\delta=2$ | $\delta=3$ | $\delta=1$ | $\delta=2$ | $\delta=3$ | $\delta=1$ | $\delta=2$ | $\delta=3$ |
| FedAvg | SimpleCNN | 10.10 | 16.70 | 29.80 | 13.10 | 23.10 | 26.10 | 10.50 | 11.10 | 15.70 |
| | ResNet-50 | 11.35 | 10.28 | 9.82 | 9.93 | 10.00 | 10.00 | 10.00 | 10.00 | 10.00 |
| FedProx | SimpleCNN | 10.10 | 12.70 | 29.90 | 13.20 | 23.20 | 26.80 | 10.60 | 10.90 | 15.90 |
| | ResNet-50 | 7.38 | 8.96 | 10.09 | 9.27 | 10.00 | 10.00 | 10.67 | 10.00 | 10.00 |
| MOON | SimpleCNN | 10.09 | 10.98 | 23.68 | 10.00 | 14.13 | 23.55 | 9.97 | 17.16 | 12.01 |
| | ResNet-50 | 10.30 | 8.55 | 11.35 | 9.55 | 10.00 | 8.97 | 11.35 | 10.00 | 10.00 |
| FedDF | SimpleCNN | 11.40 | 53.10 | 71.40 | 12.10 | 37.00 | 46.70 | 10.20 | 18.80 | 27.50 |
| | ResNet-50 | 9.76 | 34.60 | 50.99 | 10.36 | 24.38 | 41.63 | 9.92 | 22.24 | 27.43 |
| DENSE | SimpleCNN | 22.40 | 12.86 | 19.67 | 17.77 | 31.99 | 15.34 | 16.18 | 24.07 | 25.21 |
| | ResNet-50 | 10.65 | 29.26 | 37.03 | 11.49 | 15.23 | 28.38 | 11.47 | 16.67 | 24.36 |
| FedOV | SimpleCNN | 79.30 | 64.20 | 83.70 | 73.30 | 61.70 | 73.80 | 40.00 | 42.00 | 55.60 |
| | ResNet-50 | 78.69 | 70.56 | 86.99 | 66.28 | 71.70 | 75.51 | 36.25 | 50.84 | 52.50 |
| **FedTC** | ResNet-50 | **91.59** | **91.65** | **91.05** | **81.90** | **81.74** | **81.62** | **84.42** | **84.55** | **84.26** |
| | | **Non-IID Partition Strategy : LDA** | | | | | | | | |
| **Method** | **Model** | **CIFAR-10** | | | **CIFAR-100** | | | **Mini-ImageNet** | | |
| | | $\beta=0.01$ | $\beta=0.1$ | $\beta=0.5$ | $\beta=0.01$ | $\beta=0.1$ | $\beta=0.5$ | $\beta=0.01$ | $\beta=0.1$ | $\beta=0.5$ |
| FedAvg | SimpleCNN | 10.03 | 10.40 | 18.40 | 1.95 | 4.21 | 6.89 | 1.89 | 3.27 | 6.55 |
| | ResNet-50 | 10.00 | 10.00 | 10.00 | 1.00 | 1.00 | 1.00 | 1.00 | 1.00 | 1.00 |
| FedProx | SimpleCNN | 10.02 | 11.10 | 18.70 | 1.76 | 4.00 | 7.09 | 1.77 | 2.85 | 5.97 |
| | ResNet-50 | 10.00 | 10.02 | 10.00 | 1.00 | 1.00 | 1.02 | 1.09 | 0.98 | 0.84 |
| MOON | SimpleCNN | 10.09 | 17.68 | 20.41 | 1.85 | 4.06 | 6.78 | 1.79 | 3.12 | 6.22 |
| | ResNet-50 | 10.00 | 10.00 | 10.00 | 1.00 | 1.00 | 1.00 | 0.79 | 1.04 | 1.00 |
| FedDF | SimpleCNN | 12.46 | 26.30 | 35.30 | 16.23 | 20.50 | 22.40 | 4.85 | 5.45 | 6.03 |
| | ResNet-50 | 10.26 | 42.31 | 51.67 | 7.38 | 12.79 | 15.13 | 2.45 | 1.85 | 2.99 |
| DENSE | SimpleCNN | 10.00 | 46.53 | 54.35 | 12.09 | 18.69 | 29.86 | 5.20 | 5.16 | 10.58 |
| | ResNet-50 | 10.18 | 39.18 | 48.13 | 1.0 | 1.72 | 4.26 | 1.20 | 1.00 | 1.03 |
| FedOV | SimpleCNN | 46.66 | 61.70 | 65.70 | 22.13 | 29.29 | 31.20 | 13.32 | 14.36 | 12.74 |
| | ResNet-50 | 34.44 | 70.60 | 78.30 | 33.88 | 43.93 | 44.56 | 31.42 | 40.79 | 48.10 |
| **FedTC** | ResNet-50 | **84.06** | **84.69** | **84.55** | **59.69** | **59.38** | **60.42** | **73.75** | **73.50** | **74.87** |

## 4.2 MAIN RESULTS

In Tab. 1, we present the overall result of the accuracy comparison on five benchmarks, *i.e.*, MINST, Fasion-MNIST, CIFAR-10, CIFAR-100, and Mini-ImageNet, under two Non-IID partition strategies, *i.e.*, sharding and LDA. From the results, we can obtain various important conclusions.

❶ *Traditional iterative FL methods obtain terrible accuracy in one-shot setting.* Due to their reliance on multiple rounds of communication, they exhibit a substantial disparity when compared with state-of-the-art one-shot FL methods. For instance, compared with FedOV, FedAvg yields a drop as large as **59.9%** on CIFAR-10 and **37.67%** on CIFRA-100 under $\beta=0.5$. Furthermore, they even exhibit decreased accuracy with a deeper network, such as ResNet-50. This is due to the fact that deeper networks encompass a larger number of model parameters, rendering them more vulnerable to the influence of statistical heterogeneity.

❷ *FedTC yields consistent and solid improvements over competing methods.* As illustrated in Tab. 1, FedTC consistently exhibits superior performance compared to various baselines, including several state-of-the-art one-shot FL methods, across all settings. Notably, when considering the setting $\beta=0.05$, our approach showcases substantial improvements over FedOV, reaching up to **15.86%** on CIFAR-100 and **26.77%** on Mini-ImageNet. These improvements become even more significant with higher levels of statistical heterogeneity. Moreover, in comparison with other worse one-shot FL baselines like FedDF and DENSE, the improvements in the accuracy of our method are even more remarkable. The above results greatly demonstrate that our method can effectively address the statistical heterogeneity in one-shot FL.

Table 2: **Ablation studies of FedTC** on CIFAR-10 and CIFAR-100. We build up three baselines to provide more insights about our method, where **Fro.Backbone** is to freeze the pre-trained backbone, and **Det.Head** is the detached head.

| Method | Fro.Backbone | Adapter | Det.Head | CIFAR-10 | | | CIFAR-100 | | |
|---|---|---|---|---|---|---|---|---|---|
| | | | | $\beta = 0.01$ | $\beta = 0.1$ | $\beta = 0.5$ | $\beta = 0.01$ | $\beta = 0.1$ | $\beta = 0.5$ |
| $\mathcal{M}_1$ | ✓ | ✓ | | 12.57 | 30.84 | 31.73 | 6.76 | 2.38 | 9.10 |
| $\mathcal{M}_2$ | | ✓ | ✓ | 16.00 | 15.54 | 17.85 | 2.00 | 2.23 | 2.43 |
| $\mathcal{M}_3$ | ✓ | | ✓ | 76.09 | 76.16 | 76.15 | 39.47 | 38.58 | 37.20 |
| FedTC | ✓ | ✓ | ✓ | 84.06 | 84.24 | 84.55 | 59.69 | 59.38 | 60.42 |

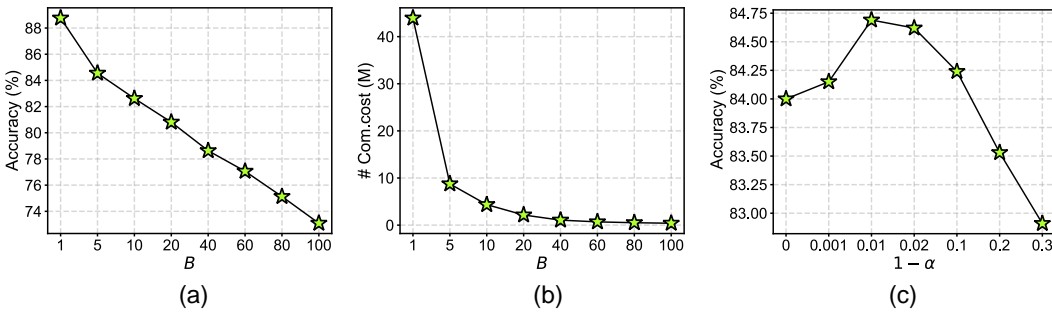

(a)         (b)         (c)

Figure 3: **The results of hyper-parameter analysis** on CIFAR-10 with $\beta = 0.1$. We illustrate the (a) test accuracy (%) and (b) communication costs versus the batch size $B$ of prototype estimation. (c) The test accuracy (%) versus the ratio of filtering, *i.e.*, $1 - \alpha$.

❸ *FedTC achieves stable performance under different heterogeneity settings.* Thanks to the ingenious design of our method, FedTC still has a stable performance though the heterogeneity of datasets is higher. In contrast, other methods yield a noticeable decrease in performance under high levels of heterogeneity. On CIFAR-10, FedOV gets a drop as **43.86%** as changing $\beta$ from 0.5 to 0.01, and the performance of DENSE is also decreased from **54.35%** to **10.00%**. This indicates that FedTC is a comprehensive solution to address heterogeneity since it effectively tackles both the backbone drift and head drift.

## 4.3 ABLATION STUDY

**Comparing with Variants.** For a deep understanding of our method, we build up three new baselines by combining different components: (1) $\mathcal{M}_1$: The adapter and head are trained on local datasets and then averaged on the server side, *i.e.*, FedAvg-v with adapter and set round to 1. (2) $\mathcal{M}_2$: FedTC with the backbone that fine-tunes the pre-trained model on local datasets through FedAvg. (3) $\mathcal{M}_3$: FedTC without adapter. We provide the results of comparison on CIFAR-10 and CIFAR-100 in Tab. 2. Apparently, the results of $\mathcal{M}_1$ and $\mathcal{M}_2$ demonstrate that training on local datasets will lead to significant drift in both the backbone and head. Averaging such biased models will result in substantial performance degradation. For example, $\mathcal{M}_1$ and $\mathcal{M}_2$ yield a drop in accuracy from **84.55%** to **17.85%** and **31.73%** on CIFAR-10 with $\beta = 0.5$, respectively. The above results further demonstrate the conclusions as stated in §3.2. By comparing $\mathcal{M}_3$ with FedTC, we can see that the adapter greatly improves the accuracy through learning the domain-specific information.

**Hyper-parameter Analysis.** FedTC involves only two hyper-parameters, *i.e.*, the ratio $\alpha$ of preserved features and the batch size $B$ to estimate the batch-level prototypes. As depicted in Fig. 3(a) and (b), larger $B$ can yield a drop in accuracy but decrease the communication costs since it reduces the number of prototypes. Fewer prototypes provide limited class-relevant information for the head and adapter, thereby degrading the performance. From Fig. 3(c), we see that it is important to choose an appropriate $\alpha$. Smaller $\alpha$ may filter out normal features while larger $\alpha$ may fail to filter out outliers.

## 4.4 COMPLEXITY ANALYSIS

**Communication Cost.** We present the communication cost of all methods with 10 clients and ResNet-50 in Tab. 3. The communication cost is the total of all clients and we only calculate the cost of transferring from client to server, ignoring the cost of downloading the pre-trained model from the public website. Consequently, the communication cost is attributed to model parameters for all methods except FedTC, where our method incurs the communication cost from the prototypes. The prototype is a $\mathbb{R}^L$ vector and $L$ is 1024, which is significantly smaller compared with the size of model parameters. From the table, we can see that the prototypes transmitted by all clients together amount to only **8.75 (MB)**, far less than the model parameters as large as **380.48 (MB)**. Such minimal communication overhead offers a significant advantage to our method in real-world applications.

Table 3: **Efficiency comparison** on CIFAR-10 with 10 clients and ResNet-50, where **# Com.cost** is the total communication cost of all clients and **# Train.time** represents the training time of per epoch.

| Method | # Com.cost | # Train.time |
|---|---|---|
| FedAvg | 380.48 M | 8.05 s |
| FedProx | 380.48 M | 9.26 s |
| MOON | 380.48 M | 10.51 s |
| FedDF | 380.48 M | 8.44 s |
| DENSE | 380.48 M | 8.10 s |
| FedOV | 380.48 M | 9.34 s |
| **FedTC** | **8.75 M** | **0.42 s** |

**Computation and Training Cost.** FedTC involves only a small amount of server training, *i.e.*, a lightweight head and adapter, and forward inference without gradient computation on the client side, thus incurring relatively small computation costs. In contrast, other methods require heavyweight training for the whole model on the client side, and DENSE and FedDF even require extensive server-side training and computation for distillation. Except for the cost of local training, the ensemble-based method, *i.e.*, FedOV, leverages all local models to predict the result for each image, incurring substantial computational costs. In Tab. 3, we compare the training time per epoch for all methods. Notably, the training time is counted in server training for our method, whereas for other methods, it is counted in client training. Obviously, the training time of our method is only **0.42 s** per epoch, significantly less than other methods, which demonstrates the lower computation cost of our server-side training.

## 4.5 VARYING BACKBONE ARCHITECTURES

FedTC provides a favourable solution for training large models, a fundamental challenge in FL. In Tab. 4, we present the experimental results of using larger pre-trained models as backbones. Clearly, larger models possess better feature representation capabilities, which is beneficial to obtaining class-related information from client data, thereby improving the classification accuracy of the detached head. For example, the performance of FedTC yields a significant improvement from **84.55%** to **97.10%** with ViT-L/14. Besides, FedTC greatly reduces communication overhead by transmitting prototypes, *e.g.*, the communication cost of using ViT-L/14 is **6.56 M**,

Table 4: **Result of FedTC with varying architectures** on CIFAR-10 with $\beta = 0.5$. **# Par.cost** and **# Pro.cost** are the communication cost of transmitting parameters and prototypes, respectively, which is a total of 10 clients.

| Architecture | # Par.cost | # Pro.cost | Acc(%) |
|---|---|---|---|
| ResNet-50 | 380.48 M | 8.75 M | 84.55 |
| ResNet-101 | 546.60 M | 4.37 M | 87.86 |
| ViT-B/16 | 832.00 M | 4.37 M | 94.06 |
| ViT-L/14 | 2191.40 M | 6.56 M | 97.10 |

only **0.29%** of the cost of transmitting parameters. The above results indicate that FedTC can be effectively applied to large models, even when deployed on edge devices.

## 5 CONCLUSION

In this work, we focus on addressing statistical heterogeneity in one-shot FL, yielding a novel one-shot FL framework, FedTC. Based on the comprehensive analysis, we utilize a fixed pre-trained backbone to capture class-relevant information by class prototypes. The prototypes are further used to train a detached head on the server side. By using prototypes as a bridge, we cleverly transform biased client-side training into unbiased server-side training. Moreover, we introduce a feature outliers filtering strategy and adapter to further improve the performance of our method. Extensive experiments demonstrate the outstanding performance of our method. Due to extremely low communication and computation costs, FedTC can be perfectly deployed in edge devices.

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

This appendix provides more details, additional experiments, and privacy analysis for our paper, which is organized as follows:

- §A shows the algorithm of FedTC and details of the adapter.
- §B presents the additional experimental details and results.
- §C provides a comprehensive analysis of the privacy security of our proposed method.

## A    MORE DETAILS OF OUR APPROACH

### A.1    DETAILED ALGORITHM

In Alg. 1, we illustrate the detailed procedure of FedTC. Before the training procedure of the server, each client employs the pre-trained backbone to perform forward inference, generating latent features for input images. Subsequently, we introduce a feature outlier filtering strategy to eliminate outliers from the set of latent features, based on similarity ranking. After that, the client estimates the batch-level prototypes and transmits them to the server. Notably, the backbone remains frozen, and there is no training procedure carried out on the client side. On the server side, we utilize these prototypes to train both the adapter and head. The adapter can further acquire domain-specific knowledge, thereby mitigating domain shifts between the pre-trained and real-world datasets.

---

**Algorithm 1:** FedTC

**Input:** $K$ local datasets: $\{D^k\}_{k=1}^K$, epochs $E$, learning rate, $\eta$, Pre-trained backbone $\boldsymbol{w}_{\mathcal{B}}$
**Output:** $\boldsymbol{w}_{\mathcal{G}}^*, \boldsymbol{w}_{\mathcal{H}}^*$

1   **Client-side Execution:**
2   **for** *client* $k = 1, 2, ..., K$ *parallelly* **do**
3      **for** $(\boldsymbol{X}_i^k, \boldsymbol{Y}_i^k) \sim D^k$ **do**
4        $\boldsymbol{Z}_i^k = \mathcal{B}^k(\boldsymbol{w}_{\mathcal{B}}, \boldsymbol{X}_i^k)$
5      **end**
6      **for** $v \sim \boldsymbol{V}$ **do**
7        $\boldsymbol{P}_v^k = \frac{1}{n_v^k} \sum_{i=1}^{n_v^k} \boldsymbol{Z}_{v,i}^k$
8        $\phi_v^k = \{\phi_{v,i}^k\}_{i=1}^{n_v^k}, \ \phi_{v,i}^k = \frac{\boldsymbol{Z}_{v,i}^k \cdot \boldsymbol{P}_v^{k\,T}}{\|\boldsymbol{Z}_{v,i}^k\| \times \|\boldsymbol{P}_v^k\|}$
9        $\widetilde{\boldsymbol{Z}}_v^k = \{\boldsymbol{Z}_{v,i}^k\}_{i \in \mathcal{Q}}, \ \mathcal{Q} = \mathbb{I}(top(\phi_v^k, \alpha))$
10       $\widetilde{\boldsymbol{P}}_{v,j}^k = \frac{1}{B} \mathbb{E}_{B_j \sim \mathcal{Q}} \sum_{i \in B_j} \widetilde{\boldsymbol{Z}}_{v,i}^k$
11       $\widetilde{\boldsymbol{P}}_v^k = \{\ldots, \widetilde{\boldsymbol{P}}_{v,j}^k, \ldots\}$
12       Transmit $\widetilde{\boldsymbol{P}}_v^k$ to server.
13      **end**
14   **end**

15   **Server-side Execution:**
     `// Client-side Execution first.`
16   Collect prototypes $\widetilde{\boldsymbol{P}}_v^k$ from clients .
     `// Construct prototype dataset.`
17   $\widetilde{D} = \{\widetilde{\boldsymbol{P}}_v, v\}_{v \in \boldsymbol{V}}, \ \widetilde{\boldsymbol{P}}_v = \{\widetilde{\boldsymbol{P}}_v^k\}_{k \in \mathcal{K}_v}$
18   Initialize $\boldsymbol{w}_{\mathcal{G}}^0, \boldsymbol{w}_{\mathcal{H}}^0$
19   **for** *epoch* $e = 1, 2, ..., E$ **do**
20      $\mathcal{L} = 0$
21      **for** $(\widetilde{\boldsymbol{P}}_i, v_i) \sim \widetilde{D}$ **do**
22        $\mathcal{L} \mathrel{+}= l_{ce}(\mathcal{G}(\boldsymbol{w}_{\mathcal{G}}, \mathcal{H}(\boldsymbol{w}_{\mathcal{H}}, \widetilde{\boldsymbol{P}}_i)), v_i)$
23      **end**
24      $\boldsymbol{w}_{\mathcal{G}}^e, \boldsymbol{w}_{\mathcal{H}}^e = (\boldsymbol{w}_{\mathcal{G}}^{e-1} \circ \boldsymbol{w}_{\mathcal{H}}^{e-1}) - \eta \nabla \mathcal{L}$
25   **end**
26   Select $\boldsymbol{w}_{\mathcal{G}}^*, \boldsymbol{w}_{\mathcal{H}}^*$.
27   Return $\boldsymbol{w}_{\mathcal{G}}^*, \boldsymbol{w}_{\mathcal{H}}^*$.

---

### A.2    DETAIL OF ADAPTER

We provide the detailed structure of the adapter in Tab. 5. The adapter comprises two fully connected (FC) layers, with ReLU and normalization operations inserted between FC layers. *Normalize* is implemented using torch.nn.functional.normalize. The adapter is lightweight yet highly effective in mitigating domain shifts, thereby substantially enhancing performance. Following the adapter, we use a linear classifier as the head to make predictions.

## B    ADDITIONAL EXPERIMENTS

### B.1    FURTHER DETAILS

In this Appendix, we show more details of our experiments in §4.

Table 5: **Detailed structure of the adapter and head.** The adapter is a lightweight module, consisting of two fully connected layers (FC). We list parameters with a sequence of input and output dimensions.

| Layer | Details |
|---|---|
| Adapter.Layer1 | FC($L$, 1024),    ReLU,    Normalize |
| Adapter.Layer2 | FC(1024, 512),    ReLU,    Normalize |
| Head | FC(512, num_class) |

**Dataset Details.** First, we illustrate the detailed information of five datasets in Tab. 6. All methods utilize the same data processing, but there are some differences in data processing between SimpleCNN and ResNet. The data processing of SimpleCNN is from (Diao et al., 2023) while the pre-trained ResNet employs the same data process of the pre-trained procedure. Besides, for MNIST and Fashion-MNIST, the 1-channel images are transformed into 3-channel images by *transforms.Grayscale* before inputting into ResNet.

**Hyper-parameters of Baselines.** There are some important hyper-parameters involved in compared baselines. For example, MOON and FedProx have a hyper-parameter $\mu$ to control the contribution of additional regulation loss. The $\mu$ of FedProx is tuned from {0.0001, 0.001, 0.01, 0.1} and empirically set to 0.001. For MOON, the $\mu$ is set to 1 and the temperature hyper-parameter is 0.5. Following FedOV (Diao et al., 2023), we utilize half of the test set as the proxy dataset for knowledge distillation of FedDF and evaluate the student model on another half of the test set. The hyper-parameters of FedOV and DENSE are set according to the description in their papers.

Table 6: **Detailed information of using datasets.**

| Property | MNIST | Fasion-MNIST | CIFAR-10 | CIFAR-100 | MINI-ImageNet |
|---|---|---|---|---|---|
| # of train samples | 60000 | 60000 | 50000 | 50000 | 50000 |
| # of test samples | 10000 | 10000 | 10000 | 10000 | 10000 |
| # of classes | 10 | 10 | 10 | 100 | 100 |
| Image size | (28, 28, 1) | (28, 28, 1) | (32, 32, 3) | (32, 32, 3) | (84, 84, 3) |

## B.2 SCALABILITY

To explore the scalability of FedTC, we evaluate the one-shot FL methods with different numbers of clients on CIFAR-10 and present the variation of test accuracy in Fig. 4. As shown in the figure, our method still yields the best accuracy when increasing the number of clients. More importantly, our approach has achieved quite stable performance under varying numbers of clients, which reveals that FedTC can be well deployed in a large federation.

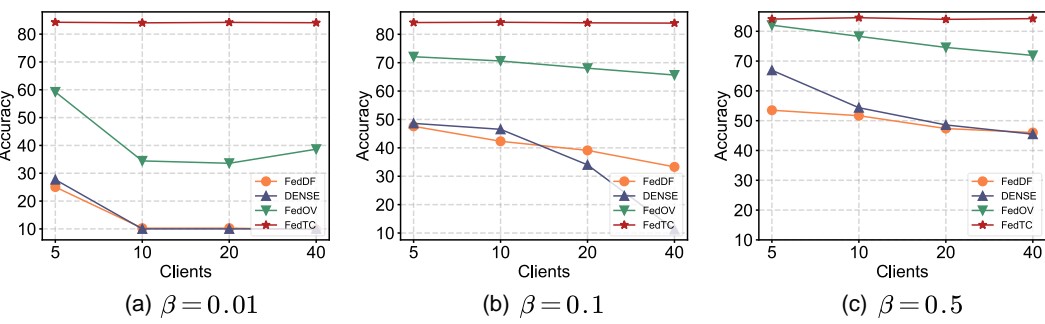

(a) $\beta = 0.01$    (b) $\beta = 0.1$    (c) $\beta = 0.5$

Figure 4: **Test accuracy (%) versus the number of clients** on CIFAR-10 with different settings.

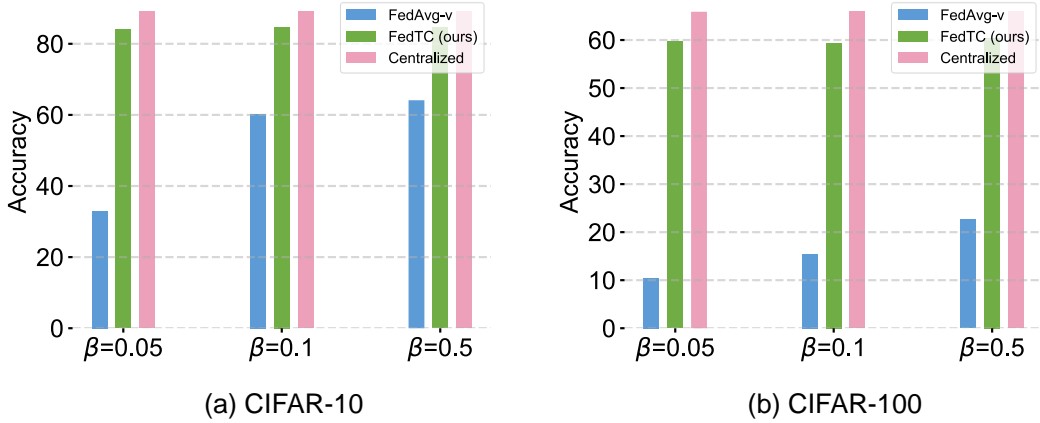

(a) CIFAR-10    (b) CIFAR-100

Figure 5: **Test accuracy (%) comparison among FedAvg-V, FedTC and centralized learning** on CIFAR-10 and CIFAR-100 with various settings.

## B.3 EFFECTIVENESS OF FEDTC FOR HEAD DRIFT

To further analyze the effectiveness of FedTC for head drift, we compared FedTC with FedAvg-V and the centralized learning method in §3.2. As depicted in the Fig. 5, FedTC yields significant improvements over FedAvg-V. Such a result suggests that parameter averaging mechanisms can still introduce significant bias even after undergoing multiple rounds of training. This bias can impact the overall performance of global updates, even when only a small number of parameters are being updated. In contrast, our detached head is not affected by statistical heterogeneity since it is updated on unbiased prototypes, thereby yielding a much higher global accuracy. Both centralized methods and FedTC utilize a pre-trained backbone while only updating the head. Therefore, the performance degradation of FedTC compared to centralized training is attributed to the compression of information by prototypes.

## B.4 ANALYSIS OF FEATURE OUTLIER FILTERING

To provide more insights about the feature outlier filtering strategy, we randomly select a client and use the T-SNE (Van der Maaten & Hinton, 2008) to visualize the latent features in Fig. 6, where the feature outliers are marked with ⋆ and ⋆. We can observe that the similarity measurement mechanism enables us to detect outliers, which are located on the boundary of the feature space. By filtering out them, we can enhance the quality of prototypes, enabling the head to learn more precise class-relevant information.

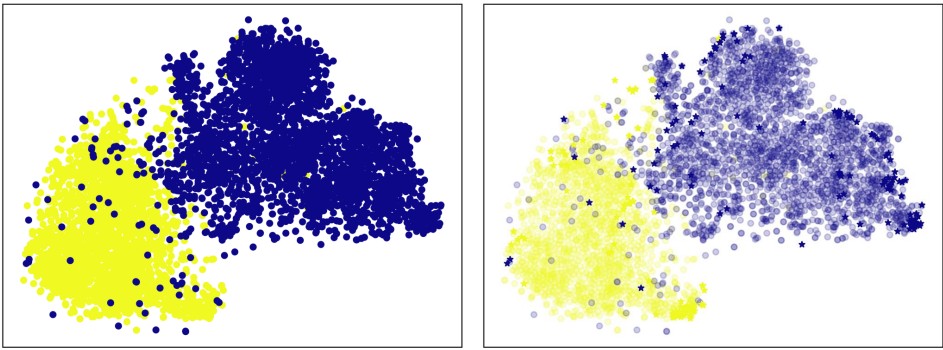

Figure 6: **T-SNE (Van der Maaten & Hinton, 2008) visualization of latent features** on CIFAR-10 with $\delta = 2$, where ● and ● are features of two classes, ⋆ and ⋆ are corresponding feature outliers. The $\alpha$ is set to 0.98.

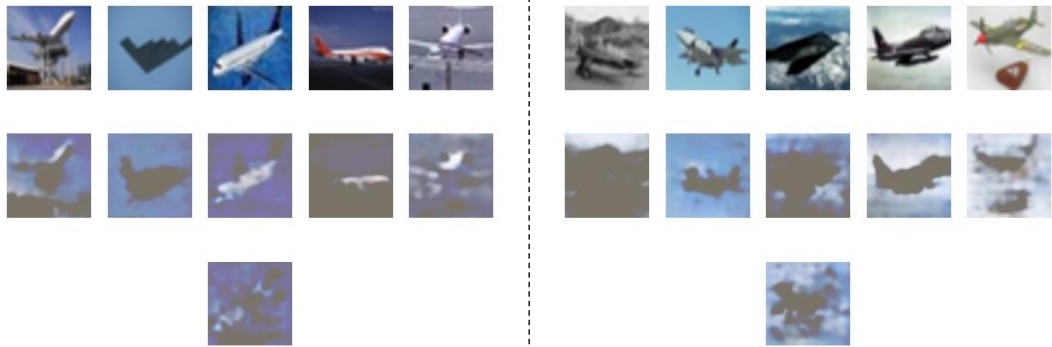

Figure 7: **The results of source-to-source reconstruction attack** on CIFAR-10. *Top*: original images. *Mid*: reconstructed images from the individual feature. *Bottom*: reconstructed images from the prototype.

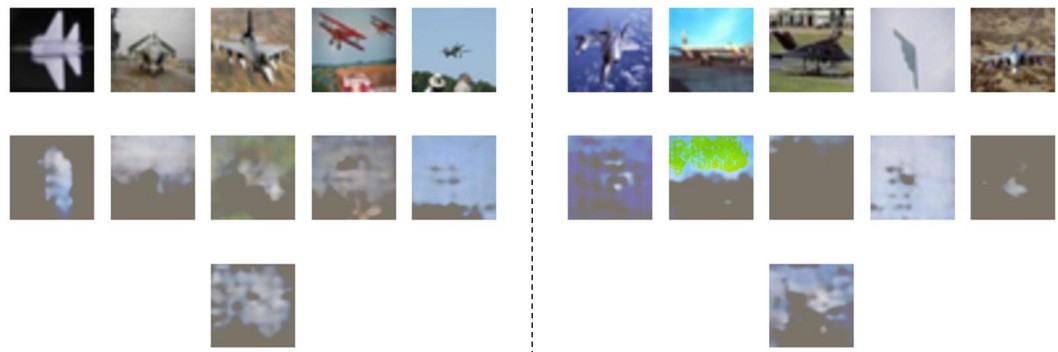

Figure 8: **The results of source-to-target reconstruction attack** on CIFAR-10. *Top*: original images. *Mid*: reconstructed images from the individual feature. *Bottom*: reconstructed images from the prototype.

## C    PRIVACY ANALYSIS

### C.1    EXPERIMENTS OF RECONSTRUCTION ATTACK

Although prototype learning is widely employed in federated learning, there remains a lack of in-depth investigations into the privacy-security aspects of prototype transfer. In this section, we conduct an exploratory experiment through the feature reconstruction attack, a type of white-box attack, to evaluate the privacy security of prototypes. We first assume there exists a malicious client (*i.e.*, denoted as source) in the federation who wants to capture the privacy of the target client. Furthermore, we have relaxed a condition where the source client can access the prototypes of the target client, even though this is impossible in our method, which strictly employs one-way transmission of prototypes between clients and a trusted server. This is different from existing works (Tan et al., 2022b;a; Huang et al., 2023), which share prototypes among different clients, thereby increasing the risk of being captured by malicious clients. After defining this scenario, we train an auto-encoder at the source client. The auto-encoder incorporates a learnable decoder after the fixed pre-trained backbone to reconstruct the latent features back into the original image. We utilize the well-trained auto-encoder to infer original images from the features and prototypes.

The results of the source-to-source attack and source-to-target attack are presented in Fig. 7 and 8, respectively. We can observe that the reconstructed images from individual features in the source-to-source reconstruction attack have similar semantic information to the original images, but the source-to-target attack does not. The above result indicates that transferring features is insecure when clients share similar data distributions. However, in terms of prototypes, the reconstructed

Table 7: **Test accuracy (%) of FedTC** with privacy-preserving techniques on CIFAR-10.

| Methods | Add Noise to | Noise Type | CIFAR-10 | | |
|---|---|---|---|---|---|
| | | | $\beta = 0.01$ | $\beta = 0.1$ | $\beta = 0.5$ |
| FedOV | - | - | 34.44 | 70.60 | 78.30 |
| **FedTC** | Prototype | $Laplace(s = 0.05,\ p = 0.1)$ | 75.04 | 75.02 | 75.63 |
| | | $Gaussian(s = 0.05,\ p = 0.1)$ | 77.77 | 77.52 | 76.97 |
| | | $Laplace(s = 0.05,\ p = 0.05)$ | 75.37 | 75.18 | 76.08 |
| | | $Gaussian(s = 0.05,\ p = 0.05)$ | 78.08 | 77.68 | 77.19 |
| **FedTC** | Image | $Laplace(s = 0.2,\ p = 0.1)$ | 73.35 | 76.08 | 75.48 |
| | | $Gaussian(s = 0.2,\ p = 0.1)$ | 73.35 | 76.08 | 75.48 |
| | | $Laplace(s = 0.2,\ p = 0.05)$ | 76.58 | 73.69 | 76.39 |
| | | $Gaussian(s = 0.2,\ p = 0.05)$ | 76.58 | 73.69 | 76.39 |

(a) Original     (b) Gaussian     (c) Laplace

Figure 9: **Illustration of (a) original images, (b) images with Gaussian noise, (c) images with Laplace noise** on CIFAR-10.

images do not contain any semantic information in both the source-to-source attack and source-to-target attack. This demonstrated that we cannot obtain user privacy from the prototypes.

### C.2 COMBING WITH DIFFERENTIAL PRIVACY

We can incorporate FedTC with privacy-preserving techniques to further enhance privacy protection. Similar to FedPCL (Tan et al., 2022b), we added varying levels of noise $\epsilon \sim (0, s)$ into prototypes and original images, respectively.

$$\widetilde{t} = t \times (1 - p) + \epsilon, \tag{6}$$

Where $t$ represents the images or prototypes, $p \in (0, 1)$ is the perturbation coefficient of the noise and $s \in (0, 1)$ is the standard deviation of the noise. From Tab. 7, we observed that FedTC still achieves a high level of accuracy with different noise. Besides, we surprisingly find that FedTC yields consistent performance despite the addition of different types of noise to the images, *i.e.*, *Laplace* and *Gaussian*. This shows that FedTC is robust to the type of noise when adding image-level noise. In Fig. 9, we show the original images and images with various noises. It can be observed that the added noise in Tab. 7 effectively preserves user privacy.

