# OpenReview forum: "One-shot Federated Learning with Training-Free Client"
_ICLR.cc/2024/Conference — ICLR 2024 Conference Withdrawn Submission_

### Official Review · Reviewer_qeaq · 2023-10-21

**Soundness:** 3 good
**Presentation:** 3 good
**Contribution:** 2 fair
**Rating:** 3
**Confidence:** 4

**Summary:**

This paper proposes a one-shot federated learning algorithm that utilizes a pre-trained model and only adapts the classifier head and an adapter of the model. On the client side, local data is used for calculating the class prototypes, and a confidence-aware approach is adopted to filter out feature outliers. On the server, the class prototypes are used for training the classifier head and adapter. Results show the proposed algorithm has great performance.

However, the proposed approach is marginal in novelty and the authors seem to ignore some most relevant existing papers which will weaken their claims and contributions.

**Strengths:**

* This paper is well-written and well-organized.
* The motivation of using pre-trained models to boost one-shot federated learning is good.
* Some insights are new.
* The experiments are mostly thorough.

**Weaknesses:**

* Lack of novelty.
  - The main idea of this paper shares a huge overlap with CCVR [1], but the author even didn't mention it. In CCVR, the authors retrain the classifier on the server using the class prototypes of clients' local data after federated learning. Ignoring the small tricks in the proposed method (e.g., filtering out feature outliers and adding an adapter), the only difference between CCVR and the proposed FedTC is that FedTC uses a pre-trained model to replace the federated training of the global model. Thus, I reckon the authors may have limited technical novelty contributions.
  - Actually, the most interesting insight of this paper is that *fixing pre-trained backbones and adapting the classifiers are enough to boost the generalization in federated learning*. The classifiers are also biased if federated learning, so the authors propose to learn the unbiased classifier via prototypes. It makes me wonder: how about initializing the prototypes as the classifiers? Since the prototypes are already expressive enough to distinguish the features into the classes' centroids. Therefore, the proposed method needs further justification.

* Lack of fair comparison.
  - Following my last point, the authors should include the most straightforward approaches into the experiment comparison to justify the method design. Also, FedAvg with only classifier training is also necessary to be a baseline, i.e., FedAvg-v in Figure 1.
  - I think it is a bit unfair for the baselines to train the backbones, and as a result, the baselines are extremely low. I think maybe a more fair comparison is to also fix the backbone and only train the classifier for the baselines (e.g., for DENSE).

* Discussion on current literature is inadequate.
  - Besides CCVR, I think the authors may miss some important related works that, at least, should be cited and discussed.
  - For the role of pre-trained models in federated learning: [2] [3].
  - For class prototypes and classifier biases in federated learning: [4] [5].

* Lack of necessary experiments.
  - Some experiments are needed. For example, experiments with more numbers of clients (also with partial participation) are needed to verify the scalability of the proposed method. It is clear that in the current version, the number of clients is only 10, which is on a small scale.

-----
[1] Luo M, Chen F, Hu D, et al. No fear of heterogeneity: Classifier calibration for federated learning with non-iid data[J]. Advances in Neural Information Processing Systems, 2021, 34: 5972-5984.

[2] Chen H Y, Tu C H, Li Z, et al. On the importance and applicability of pre-training for federated learning[C]//The Eleventh International Conference on Learning Representations. 2023.

[3] Nguyen J, Wang J, Malik K, et al. Where to begin? on the impact of pre-training and initialization in federated learning[C]//The Eleventh International Conference on Learning Representations. 2023.

[4] Li Z, Shang X, He R, et al. No Fear of Classifier Biases: Neural Collapse Inspired Federated Learning with Synthetic and Fixed Classifier[C]//Proceedings of the IEEE/CVF International Conference on Computer Vision (ICCV). 2023.

[5] Dai Y, Chen Z, Li J, et al. Tackling data heterogeneity in federated learning with class prototypes[C]//Proceedings of the AAAI Conference on Artificial Intelligence. 2023, 37(6): 7314-7322.

**Questions:**

See the weakness part above.

---

### Official Review · Reviewer_E58J · 2023-10-22

**Soundness:** 2 fair
**Presentation:** 2 fair
**Contribution:** 3 good
**Rating:** 5
**Confidence:** 4

**Summary:**

This paper introduces a novel one-shot federated learning (FL) algorithm, termed FedTC, tailored for settings with high statistical heterogeneity. At first, the authors observe that updating local models on biased datasets results in representation differences (referred to as 'backbone shift'), and averaging only on the local heads leads to a substantial performance gap compared to centralized learning (termed 'head drift'). Therefore, the authors propose a frozen and pre-trained backbone for the clients and a detached head for the server to avoid the drift issue. Specifically, inspired by prototype learning, unbiased prototypes are computed and assembled from all clients. An adapter is then employed to help the server-side head learn domain-specific knowledge from these prototypes. Experiments demonstrate the effectiveness and efficiency of the proposed method, FedTC.

The main contribution is that this work introduces pre-trained models and prototype learning into one-shot FL, which limits communication to a single round, attempting to retain FL performance under high statistical heterogeneity.

**Strengths:**

- Addressing high statistical heterogeneity in one-shot FL is imperative and demands immediate attention.
- The proposed method is novel to me. Utilizing a pre-trained backbone combined with prototype learning to generate class prototypes for aggregation to get an unbiased dataset is novel for overcoming client drift in one-shot FL. The strategy of having a fixed local pre-trained backbone on the client side also reduces the computation cost for edge devices, which is practical in real-world scenarios.
- The paper is easy to follow.

**Weaknesses:**

- Observations on the backbone drift and head drift don’t seem to provide new insights. Training for a large number of epochs on local clients can lead to overfitting, and local models might diverge significantly from each other in a non-iid setting, as already demonstrated in [1-2].
    - It would have been better to show the similarity in feature representations across different layers.
    - It would be more convincing to choose froze if the proposed method FedTC performs worse with few local epochs (e.g., 10) for updating the backbone or subsequent layers of the backbone (not the $M_1$ case in the paper; include the prototype learning as well).

- Unclear experimental details and an unfair comparison.
    - In the ablation studies, it would be better to show how the proposed method perform without the prototypes part (simply aggregation).
    - Also, this paper chooses pre-trained weights from CLIP and pre-trained weights from ImageNet for different scale datasets. It would provide better clarity if the authors could show how the pre-trained weights would affect the performance of the proposed method, FedTC.
    - All methods for comparison are employed with the adapter, but for example, the DENSE method uploads local models to the server; in this case, the adapter might not be necessary and even could result in a performance drop.
    - Clarification is needed on whether the competing methods use identical pre-trained weights. Moreover, the DENSE method is a data-free methodology; how is this method implemented for comparison?

- Currently, experiments only consider local models that are homogeneous, while it might have different model architectures across clients in the real world [3]. It might be beneficial if the proposed method could further consider different local models with different pre-trained weights and apply an adapter to handle prototypes in different dimensions.

[1] Lin, Tao, et al. "Ensemble distillation for robust model fusion in federated learning." Advances in Neural Information Processing Systems 33 (2020): 2351-2363.

[2] Heinbaugh, Clare Elizabeth, Emilio Luz-Ricca, and Huajie Shao. "Data-Free One-Shot Federated Learning Under Very High Statistical Heterogeneity." The Eleventh International Conference on Learning Representations. 2022.

[3] Zhang, Jie, et al. "Dense: Data-free one-shot federated learning." Advances in Neural Information Processing Systems 35 (2022): 21414-21428.

**Questions:**

Please see the weakness above.

The adapter is introduced ahead of the detached head, so in Eq. (5), it might be $\mathcal{H}(w_h, \mathcal{g}(w_g, p))$.

---

### Official Review · Reviewer_y2sF · 2023-10-31

**Soundness:** 3 good
**Presentation:** 3 good
**Contribution:** 2 fair
**Rating:** 3
**Confidence:** 4

**Summary:**

This paper introduces FedTC (Training-free Clients), a federated learning approach that requires no training on the clients. The core idea behind FedTC is to keep the entire framework's backbone fixed, allowing clients to extract representative features of their images using the common backbone and upload these features to the server. The server then employs these features to train adapters and classification heads appended to the backbone, thereby completing the training of the global model.

**Strengths:**

1. The paper is easy to follow and has a clear line of thought.
2. The appendix contains enough discussions on privacy issues.

**Weaknesses:**

1. FedTC's inability to train the backbone and only making minor adjustments to pre-trained backbones on specific datasets significantly limit its applications.
2. The motivation in this paper does not emphasize the issue of client computation but focuses on addressing client drift. This may not align well with FedTC's main advantage, which is training-free clients, rather than generalization performance.
3. Most existing federated learning methods should perform reasonably well on the selected small datasets, especially when everyone shares the same pre-trained backbone. However, as observed in Table 1, most methods lack classification capability. These results may not hold much meaning. This paper should compare these non-one-shot methods after multiple iterations and expand the comparison to more one-shot methods.

**Questions:**

See weaknesses

---

### Official Review · Reviewer_7xGT · 2023-11-05

**Soundness:** 2 fair
**Presentation:** 3 good
**Contribution:** 2 fair
**Rating:** 5
**Confidence:** 2

**Summary:**

This paper proposed a method to tackle the task of  one-shot federated learning under the setting of non IID. The main idea seems to divides the model into backbone and head, deploying them separately on the client and server sides.

In the setting of with no sufficiently large training data available, it's know that only fine tuning the head and freezing the backbone can lead to better results, which was reported in object detection. Thus I am not supervised to see that freezing the backbone and only training the head works better in federated learning.

**Strengths:**

The ides is simple and I believe it should work, as I said above, though I am not sure if this can be a good contribution to federated learning as I am not closely following recent federated learning literature.
Experiments show this simple idea works well.

**Weaknesses:**

Experiments are not well designed.
1) I think the authors should first compare FedTC against recent one-shot federated learning methods on standard IID settings. FedTC doesn't need to beat those methods, but I'd expect FedTC show on par results compared with those methods. I did not see that in the paper.
2) Table 1 which is the main results of the paper, shows FedTC largely outperforms other methods. But to my understanding all of the compared methods (maybe most, I did not check carefully) are not designed to handle non IID cases. Is it expected to see those methods fail?
Then what is the main point of Table 1?
3) Since the main selling point is FedTC works better on non IID datasets, I'd like to see FedTC compared against traditional non IID methods (by traditional, I mean training in the standard way rather than federated learning, which presumably should work better than federated learning)

4) Since one shot federated learning is largely unsolved, maybe it makes more sense to improve that rather than to solve much more challenging one shot, non IID, federated learning. This is just a comment.

**Questions:**

see above

---

### Official Review · Reviewer_fiUw · 2023-11-09

**Soundness:** 2 fair
**Presentation:** 3 good
**Contribution:** 2 fair
**Rating:** 5
**Confidence:** 4

**Summary:**

This paper addresses statistical heterogeneity in one-shot federated learning (FL) and proposes a novel one-shot FL framework called FedTC. This method does not require any training on the client side because it uses a pre-trained backbone. Experimental results show that FedTC outperforms standard FL methods by a significant margin. The proposed method is easy to implement but lacks novelty.

**Strengths:**

This paper addresses statistical heterogeneity in one-shot FL, yielding a novel one-shot FL framework called FedTC. Experimental results are promising when compared to standard FL methods.

**Weaknesses:**

Lack of novelty: This paper uses a straightforward approach for computing batch-wise class prototypes using a pretrained backbone.  The proposed methodology is not well explained and presented, particularly the training process of the classification head using prototypes. Additionally, the lack of elaboration and clarity in this aspect is a drawback. It would be interesting to see a more challenging experimental setup involving domain shift, such as training the pretrained CLIP model on a medical dataset or training a model that is pretrained on synthetic instances on real-world images.

**Questions:**

1. Since the pretrained backbone is not adapted to local data, the performance of the model is solely dependent on the pretrained model, which is a drawback of this method. Have you examined such a scenario with pretrained models from different timestamps during the training steps?
2.  What does \mathcal{H} represent in Eq. 5?
3. How to predict on client dataset?
4. In Algorithm 1, \mat{w}^∗_G , \mat{w}^∗_H is not defined before use in line no. 26. how do you define \mat{w}^∗_G , \mat{w}^∗_H?